# Feasibility and Acceptability of Bright IDEAS-Young Adults: A Problem-Solving Skills Training Intervention

**DOI:** 10.3390/cancers14133124

**Published:** 2022-06-25

**Authors:** Adrienne S. Viola, Gary Kwok, Kristine Levonyan-Radloff, Sharon L. Manne, Robert B. Noll, Sean Phipps, Olle Jane Z. Sahler, Katie A. Devine

**Affiliations:** 1Department of Pediatrics, Robert Wood Johnson Medical School, Rutgers Cancer Institute of New Jersey, New Brunswick, NJ 08903, USA; asv51@rwjms.rutgers.edu (A.S.V.); gkk28@cinj.rutgers.edu (G.K.); kl747@cinj.rutgers.edu (K.L.-R.); 2Department of Medicine, Robert Wood Johnson Medical School, Rutgers Cancer Institute of New Jersey, New Brunswick, NJ 08903, USA; mannesl@cinj.rutgers.edu; 3Department of Pediatrics, University of Pittsburgh, Pittsburgh, PA 15213, USA; rbn1@pitt.edu; 4Psychology Department, St. Jude Children’s Research Hospital, Memphis, TN 38105, USA; sean.phipps@stjude.org; 5Departments of Pediatrics, Psychiatry, Health Humanities & Bioethics, and Oncology, University of Rochester Medical Center, Rochester, NY 14642, USA; oj_sahler@urmc.rochester.edu

**Keywords:** Bright IDEAS, problem-solving skills, behavioral intervention, young adult, cancer, distress

## Abstract

**Simple Summary:**

Young adults with cancer face many different stressors due to a diagnosis of cancer during a unique developmental period. Interventions are needed to address their needs and help better manage distress. Bright IDEAS is a problem-solving skills-training program that has shown to improve people’s problem-solving abilities and reduce the negative affect on caregivers of children with cancer. This study aimed to evaluate if an adapted version of Bright IDEAS was feasible and acceptable to young adults with cancer. Forty young adults recently diagnosed with cancer were enrolled. The results suggested that young adults were satisfied with Bright IDEAS and supported the potential impact to improve problem-solving skills and reduce symptoms of depression and anxiety.

**Abstract:**

Background: Young adults with cancer are a vulnerable group with unique emotional, social, and practical needs. There is a lack of evidence-based interventions to address their needs and to foster skills that could increase their capacity to cope. Bright IDEAS is a problem-solving skills training intervention that has demonstrated efficacy in improving people’s problem-solving ability and reducing distress among caregivers of children with cancer. This study evaluated the feasibility and acceptability of Bright IDEAS adapted for young adults (Bright IDEAS-YA). Methods: Forty young adults recently diagnosed with cancer were enrolled in a single arm feasibility study. Results: Feasibility was demonstrated by the adequate enrollment (67.8%), retention (80.0%), and participants’ adherence to the intervention (average of 5.2 out of 6 sessions completed). Participants reported satisfaction with the intervention. Qualitative feedback identified the systematic approach to problem-solving and interaction with the trainer as strengths of the intervention. Participants demonstrated improvements in problem-solving skills and symptoms of depression and anxiety. Conclusions: In conclusion, the results support the feasibility of the intervention and an adequately powered randomized controlled trial is needed to determine the efficacy of the intervention on psychosocial outcomes.

## 1. Introduction

Young adults (YAs) with cancer diagnosed between the ages of 18 to 29 have been recognized as a vulnerable group with unique emotional, social, and practical needs due to the intersection of cancer treatment and normal developmental processes. The age range of 18–29 years has been coined ‘emerging adulthood” because it is characterized by changes in identity and instability across multiple life domains [1]. Although a cancer diagnosis at any age can be stressful, its diagnosis and treatment during the critical period of young adulthood is particularly challenging [2]. A cancer diagnosis during this time can disrupt the process of achieving desired developmental tasks in all life domains, including identity formation, education, career, financial independence, relationships, and starting a family [3,4,5]. Numerous studies have documented that young adults with cancer express needs for informational (e.g., information about illness, treatment, risk for recurrence, infertility, or age-appropriate internet sites), practical (e.g., help with insurance, child care, transportation, infertility services, or complementary and alternative medicine), and emotional (e.g., community support resources, age-appropriate camps, mental health counseling, family counseling, or spiritual counseling) support services that are often unmet [6,7,8,9,10,11,12]. Unmet psychosocial needs are associated with poorer outcomes, including greater distress and a poorer health-related quality of life [2,4,8,11,13]. YA cancer patients also demonstrated a lower quality of life compared to their peers in the general population, as well as increased levels of anxiety and depression following a cancer diagnosis [14]. Despite the documented needs of this group, there is a lack of evidence-based interventions to address the unique concerns of the young adult population [15]. In particular, there are few skills-based interventions that target young adults newly diagnosed with cancer [16]. To our knowledge, only one such intervention focused on promoting resilience among newly diagnosed adolescent and young adult patients (aged 12–25 years) in a pediatric setting [17,18]. Results showed improvements in resilience, cancer-specific quality of life, and distress, but may not translate to adult oncology settings or older young adults, given that the sample was from a pediatric oncology setting with the majority of participants being under 18 years of age [16].

One potential solution to address the individual needs of young adults with cancer is a behavioral intervention that would provide skills to manage the diverse and numerous stressors associated with a cancer diagnosis in the context of young adult life transitions, address the developing problem-solving abilities typical of this life stage, and would be relatively simple to learn and use during the highly stressful time following a diagnosis of cancer. Problem-solving skills training is an application of problem-solving therapy, which has accumulated a large body of evidence demonstrating its success at improving people’s problem-solving ability, reduces negative affect, and improves health-related quality of life [19]. Bright IDEAS is a problem-solving skills training intervention that has demonstrated efficacy in enhancing problem-solving ability and reducing negative affect in multiple randomized controlled trials with parents of children with cancer [20,21,22]. Bright IDEAS has been successfully adapted for caregivers of children with sickle cell disease, mothers of children with autism spectrum disorder, and adult cancer survivors [20,21,22,23,24,25]. Prior work shows that younger caregivers and Spanish-speaking caregivers with fewer resources benefitted more from the Bright IDEAS intervention [19]. Since Bright IDEAS is a framework that can be readily adapted to various problems, we believed it could be readily adapted to meet the needs of YA patients who tend to have less life experience in managing the demands of a serious illness. In this study, we adapted Bright IDEAS for young adults with cancer to meet their needs during the critical months following a new cancer diagnosis. The goal of this study was to evaluate the feasibility and acceptability of Bright IDEAS-Young Adults (Bright IDEAS-YA). The focus was on assessing recruitment rates and retention capability, examining YA engagement with the intervention, and determining the acceptability of the intervention in this group [26]. A secondary aim was to explore intervention effects on problem-solving ability (i.e., the targeted skill), distress, and health-related quality of life.

## 2. Materials and Methods

### 2.1. Sample

Young adults ages 18–29 who were diagnosed with cancer and on active treatment were recruited from the Rutgers Cancer Institute of New Jersey from 31 January 2018 to 24 October 2019. Potential participants were excluded if they were non-English speaking, not cognitively able to complete survey measures independently, and at the time of recruitment were experiencing a medical crisis or were deemed to have less than 6 months of life expectancy as per the physician report. Eligible participants were identified from the medical records of oncology visits for physicians treating any type of cancer (e.g., hematological, breast, testicular, cervical, colon, gynecological, and prostate) as well as through referrals from treating physicians, nurses, and social workers. Physicians were contacted to confirm potential participants met eligibility criteria and request permission to approach prospective participants.

### 2.2. Intervention

Bright IDEAS is an eight-session evidence-based, manualized, problem-solving skills intervention. We modified Bright IDEAS-YA to six sessions, which was considered to be an adequate dose in prior studies [22,27]. We also changed the materials to be simpler/more appealing, added examples relevant to young adults (e.g., changing a caregiver example of having questions about their child’s diagnosis to a young adult having questions about their own diagnosis), added psychoeducation regarding changeable vs. unchangeable aspects of problems, and expanded the use of the word “problems” to include “problems”, “challenges”, or “goals” to capture the range of issues young adults might experience. A qualified trainer (i.e., graduate student, master’s level clinician, or licensed clinician) met individually with the participant to teach them the Bright IDEAS stepwise approach to problem-solving. Participants received a user manual with worksheets to facilitate the implementation of the approach. The acronym “Bright IDEAS” stands for the key steps of the intervention. “Bright” refers to fostering a sense of empowerment and optimism that the participant is able to resolve challenges they face. Each letter in “IDEAS” stands for a step of the approach (i.e., Identify the problem, Define your options, Evaluate options, Act, and See if it worked). Session 1 focused on rapport building, teaching the Bright IDEAS model, and identifying problems or challenges that the participant would like to work on. In sessions 2–5, the trainer guided the participant through solving their own problems using the worksheets for each step. Session 6 involved a review of progress and discussion of strategies to continue to use the Bright IDEAS method instead of lapsing into ineffective problem-solving approaches. Sessions lasted about 45 min and were conducted either in person or by telephone/videoconference according to participant preference. Sessions were audio recorded for trainer supervision and to conduct a treatment integrity assessment. A checklist was used to evaluate the content and process of the sessions to ensure intervention fidelity.

### 2.3. Procedures

Eligible patients consented in person during a routine clinic visit in either the adult or pediatric oncology clinic. After providing informed consent, participants completed the baseline survey. Next, they were assigned a Bright IDEAS trainer, who reached out to schedule the first intervention session. Sessions typically took place every one to two weeks, generally in concordance with the patient’s medical visits. In-person sessions could be replaced by a phone session if a patient was unable to make it to the clinic or did not have an appointment. Participants were asked to complete the post-intervention survey immediately following their final intervention session.

### 2.4. Measures

#### 2.4.1. Feasibility and Acceptability 

Feasibility was assessed using study enrollment rates, retention rates, reasons for study dropout, and engagement with the intervention. Acceptability was assessed by satisfaction with the program using an adapted version of the Multi-Dimensional Treatment Satisfaction Measure [28]. This five-item survey asked about the usefulness of the Bright IDEAS acronym in remembering the steps of the program, the extent to which participants saw themselves continuing to use Bright IDEAS, whether they explained it to another person, whether they used it to solve a problem, and how useful they believed it would be for others on a rating scale from 1 (strongly disagree) to 5 (strongly agree). After the first 10 participants completed the intervention, four new items were added to assess the overall perceived usefulness of Bright IDEAS, the worksheets, and the participant manual, as well as the ease of talking with the trainer. Three open-ended questions asked the participants what they liked best about the intervention, what they liked least, and any suggestions for improvements. Participants were also asked these questions verbally during the final session.

#### 2.4.2. Patient-Reported Outcomes

Demographic variables (i.e., age, race/ethnicity, employment/school status, marital status) were collected at baseline. All other measures were provided at baseline and immediately following the completion of the intervention.

Social Problem-Solving Inventory-Revised Short Form (SPSI-R:S): The SPSI-R is a 25-item measure of five theoretically important constructs of social-problem solving, including positive problem orientation, negative problem orientation, rational problem-solving style, impulsive/carelessness style, and avoidance style [29]. Scores are computed for each scale as an average, with negative constructs recoded such that higher scores indicate better functioning. The total score is a sum of the subscales; it ranges from 0 to 20 with higher scores indicating a better problem-solving ability. The SPSI-R:S is characterized by strong reliability and validity estimates. In this study, the SPSI-R:S demonstrated adequate internal reliability for the total score (α = 0.86) and for all subscales except impulsive/carelessness (α = 0.38; all other α = 0.68 to 0.85).

Distress Thermometer and Problems Inventory: The Distress Thermometer is a widely used brief screening tool for distress among oncology patients [30]. We utilized an adapted version that has a distress rating from 0 to 10 and a problem checklist modified to include issues unique to AYA (e.g., concerns about parents or being isolated from friends) [31].

PROMIS Anxiety and PROMIS Depression Short Form: These 8-item short forms are widely used and validated measures of the symptoms of anxiety and depression [32]. Respondents report symptoms on a 5-point scale from never to always, with higher scores indicating higher levels of negative affect. All PROMIS short-form measures can be scored using a T-score metric, with a mean of 50 and standard deviation of 10. In this study, these measures demonstrated adequate internal reliability (α = 0.93 for anxiety, α = 0.95 for depression).

The PedsQL Generic Core [33,34] Young Adult version was used to measure health-related quality of life (HRQOL). The PedsQL generic core is a 23-item measure that yields a total score, 4 subscale scores (Physical, Emotional, Social, and Cognitive), and 2 summary scores—(Physical Health Summary and Psychosocial Health Summary). Scores are transformed on a 0–100 scale, with higher scores indicating better functioning. The PedsQL is a widely used and well-validated measure of QOL, including with AYA with cancer [4,35]. In this study, we focused on the total score, which demonstrated adequate internal reliability (α = 0.92).

### 2.5. Analysis

Statistical analyses were conducted using SPSS version 28. Feasibility and satisfaction were evaluated using descriptive analyses (means, standard deviations, frequencies). Potential differences between study participants and non-participants (those who opted to not be in the study) were evaluated with *t*-tests or Chi-square analyses. Intervention effects were explored using paired t-tests to assess the changes in outcomes between the two time points. Cohen’s d effect size was calculated, with an effect size of 0.2 considered small, 0.5 considered medium, and 0.8 considered large [36].

Responses to the open-ended questions in the follow-up survey and final intervention session were reviewed to thematically identify the strengths and weaknesses of the intervention, as well as ideas to improve the intervention.

## 3. Results

### 3.1. Feasibility

#### 3.1.1. Enrollment and Retention 

In total, 62 patients were approached with 59 being eligible for the study (3 participants were excluded as non-English speaking). Of those eligible, 40 (67.8%) completed the baseline assessment and started the intervention. Reasons for declining to participate included not having enough time or feeling overwhelmed (*n* = 6), not interested (*n* = 5), perceived adequate support/no need (*n* = 6), transferring care (*n* = 1), or being too ill (*n* = 1). There were no significant differences between participants and non-participants in age, sex, race, ethnicity, diagnosis type (blood cancer vs. other), or time since diagnosis (ps > 0.05). Participant characteristics are presented in Table 1.

Of those who started the study, 32 (80%) completed all of the study tasks (See Figure 1 for CONSORT flow diagram). The most common time of drop out was after the first session (*n* = 4), followed by after the second session (*n* = 2). Participants withdrew from the study because of worsening health (*n* = 5), general medical noncompliance (*n* = 2), and being too busy to schedule meetings with the trainer (*n* = 1). Males were more likely to withdraw from the study compared with females (35% vs. 5%, X^2^(1) = 5.63, *p* = 0.02). Participants who enrolled after a longer time from their initial diagnosis were also more likely to withdraw, F(1, 38) = 5.94, *p* = 0.02. There were no significant differences in withdrawal rates by age, race (White vs. other), ethnicity (Hispanic vs. non-Hispanic), education, employment status, or diagnosis type (blood cancer vs. other).

#### 3.1.2. Participant Engagement and Satisfaction 

Participants completed an average of 5.2 (SD = 1.8) out of 6 sessions with the trainer (range = 0 to 6). Sessions lasted an average of 46.6 min each (SD = 13.3) and were completed over an average of 77.9 days (SD = 33.0) or 11.1 weeks. Participants worked on problems including (1) coping with the uncertainty of cancer; (2) deciding to go back to school; (3) returning to work/figuring out career; (4) feeling isolated from friends; (5) eating healthy/losing weight; (6) dating with cancer; and (7) challenges moving back home with parents.

Participant satisfaction ratings are reported in Table 2. Of those who completed the program, participants reported high satisfaction with the program overall (*M* = 4.7, SD = 0.48, scale 1–5). The majority (83.8%) reported that they could see themselves using the Bright IDEAS system of problem solving in the future, and 84.4% thought that the Bright IDEAS system could be useful for their friends and family. Participants highly rated the ease of talking with their trainer (100% strongly agree or agree), followed by the participant manual (86.4%) and worksheets (81.8%).

Qualitative feedback from the survey and verbal feedback in the final session indicated that participants liked the systematic approach of Bright IDEAS and talking with a supportive trainer (Table 3). There were fewer comments in response to what they liked least; aspects commented upon were difficulties in scheduling the sessions due to treatment, finding the clinical setting stressful, and “homework.” Suggestions for improvements included adding a group or peer community component and changing the scheduling of sessions. 

### 3.2. Secondary Outcomes

#### 3.2.1. Problem-Solving Skills

Overall their problem-solving ability improved from baseline to post-intervention, representing a medium effect (d = 0.54). Subscale scores demonstrated improvements from baseline to post-intervention with small to medium effect sizes (see Table 4).

#### 3.2.2. Anxiety

Anxiety T-scores decreased from baseline to post-intervention, representing a small effect size, *d* = −0.26 (Table 4). 

#### 3.2.3. Depression

Depression T-scores decreased from baseline to post-intervention, representing a small effect size, *d* = −0.36 (Table 4).

#### 3.2.4. Quality of Life

Total health-related quality of life improved slightly, representing a small effect size, *d* = 0.16 (see Table 4).

#### 3.2.5. Distress

Participants’ distress reduced from an average of 3.16 (SD = 2.81) at baseline to 2.06 (SD = 2.16) at post-intervention, representing a small to medium effect (d = −0.47; Table 4).

## 4. Discussion

Results indicated that the Bright IDEAS-YA problem-solving skills training intervention was feasible and acceptable to young adults diagnosed with cancer. The feasibility of this study was demonstrated by adequate enrollment (67.8%), retention (80.0%), and adherence to the intervention (5.2 out of 6 sessions completed). No a priori benchmarks were established, but the enrollment and retention rates were similar to the prior study of adolescents and young adults newly diagnosed with cancer in a pediatric setting (71% enrollment, 80% retention) [16]. Withdrawal from the study was predominantly due to worsening health and feeling too ill to complete the sessions. More males than females withdrew from the study, but it is hard to draw conclusions regarding the reason for differential withdrawal given the small sample size and that the primary reason for withdrawal was worsening health/feeling too ill. However, future trials should closely monitor whether there is a differential response to the intervention by participant sex. Individuals for whom more time had elapsed since diagnosis were also more likely to withdraw. Again, caution is taken in interpreting these data due to the small sample size and reasons given for withdrawal; however, there was some qualitative suggestion to deliver the intervention closer to diagnosis and a future trial may restrict enrollment to the first several months following the diagnosis. 

Participants reported high overall satisfaction with the intervention and particularly liked the systematic approach to problem-solving and interaction with the trainer. Consistent with the broader literature, [37,38] young adults in our study reported a number of challenges during treatment for cancer, including coping with the uncertainty of cancer, challenges with work/careers or returning to school, social difficulties with friends or romantic relationships, engaging in healthy behaviors, and losing autonomy due to moving back under the care of parents. Participants were able to apply the Bright IDEAS problem-solving framework to these broad and varied stressors across multiple life domains. Given the breadth of the problems discussed and the overall indicators of engagement and satisfaction with the interventions, it appears that Bright IDEAS-YA was an acceptable approach to address challenges across the age spectrum of emerging adulthood.

The results suggest that YA can improve their problem-solving ability over the course of a short intervention as demonstrated by the medium magnitude of effect on overall problem-solving ability as well as small to medium size effects across all the positive and negative subscale domains. The largest improvement was seen within the rational problem-solving style, which is well aligned with purposefully using the Bright IDEAS steps to solve a problem. These findings are consistent with the prior literature demonstrating that brief problem-solving skills training can be effective for youth [39] as well as adult populations [40]. The findings also suggest small to medium effects on symptoms of anxiety, depression, and overall distress, consistent with previous reports of problem-solving skills training [20,21,22]. There was only a very small effect on health-related quality of life; given that participants were undergoing curative cancer treatment during the study, it can be difficult to interpret changes in HRQOL over such a short time period.

The average time to completion of the six sessions was eleven weeks. In combination with the qualitative feedback that scheduling was sometimes difficult due to treatment, flexibility in scheduling intervention sessions is needed for this population. This fits well with the broader literature recognizing the competing demands experienced by YAs during cancer care and the difficulty in recruiting them for research trials [41,42]. Further qualitative feedback indicated a desire for peer support; while this is well-aligned with adolescent and young adult preferences for connecting with peer survivors, [43] it is often difficult to implement in practice as YA patients tend to live geographically distant from their treating cancer center and have many competing demands. Technology may overcome some of these barriers to harnessing peer support [44]. However, engagement even in online communities can be variable [45].

Despite the many strengths of this intervention, this study had several limitations associated with its primary purpose focusing on feasibility and acceptability. The small sample size was appropriate for assessing feasibility as the primary aim, and as such there was no a priori power calculation for the secondary outcomes. As a single arm, non-randomized study, there was no comparison group and we cannot rule out that improvements in secondary outcomes were due to the passage of time rather than the intervention. The sample was drawn from a single NCI-designated comprehensive cancer center and thus representative of the local catchment area, which may not generalize broadly to other community settings. Finally, the sample was heterogeneous with participants in various stages of their illness. While the median time since diagnosis was seven weeks, the range was large (up to 86 weeks) and this could influence outcomes. An adequately powered randomized trial is needed to address some of these limitations. Additionally, future studies should consider factors that may influence responses to the intervention, such as participant sex, baseline levels of distress, or unmet needs. Future studies should also carefully plan recruitment and retention strategies that acknowledge the preferences of young adults as well as the barriers to participation, such as competing time commitments, the burden of committing to a research study, and the interest/preference for the types of study activities and incentives offers [46].

## 5. Conclusions

In conclusion, the Bright IDEAS-YA intervention was both feasible and acceptable to young adults diagnosed with cancer. In addition, our results indicate that Bright IDEAS-YA improves the behavioral target of problem-solving skills and may reduce symptoms of depression, anxiety, and distress. These promising results warrant further study in an adequately powered, randomized controlled trial to examine the efficacy of the intervention on improving psychosocial outcomes.

## Figures and Tables

**Figure 1 cancers-14-03124-f001:**
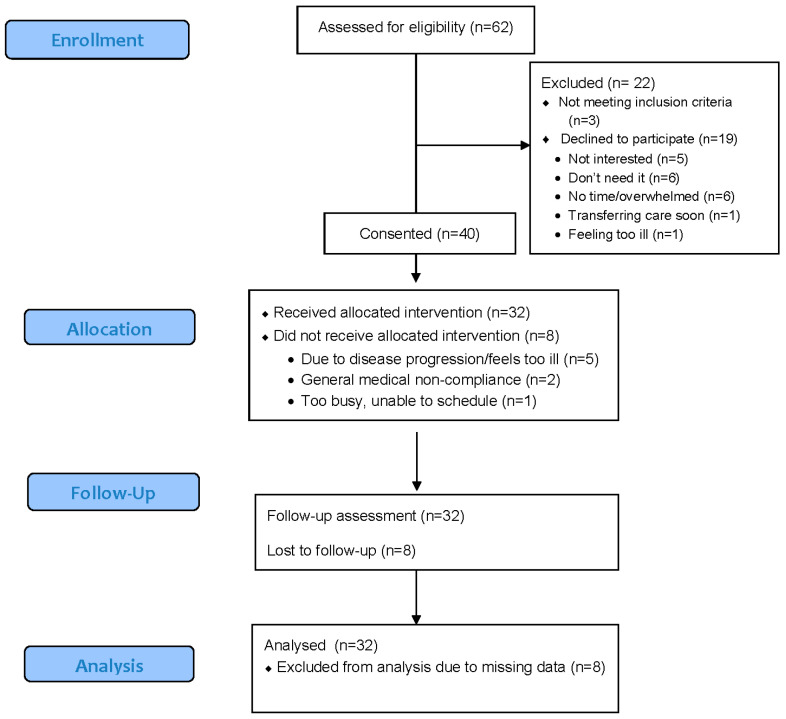
CONSORT flow diagram.

**Table 1 cancers-14-03124-t001:** Participant Characteristics (*n* = 40).

Characteristic	*M* (SD) or *n* (%)
Current Age in Years, *M*(SD)	23.9 (3.3)
Range	18–29
Female, *n* (%)	20 (50%)
Race, *n* (%)	
White	18 (45.0%)
Black	5 (12.5%)
Asian	4 (10.0%)
More than one race	1 (2.5%)
Other/Unknown/missing	12 (30.0%)
Hispanic Ethnicity, *n* (%)	17 (42.5%)
Single/Never Married, *n* (%)	36 (90.0%)
Employment Status, *n* (%)	
Working Full-Time	22 (55.0%)
Full-Time Student	5 (12.5%)
Employed Part-Time/Student	6 (15.0%)
Unemployed	6 (15.0%)
Homemaker/Caregiver	1 (2.5%)
Highest Grade Completed, *n* (%)	
Less than HS	1 (2.5%)
High School/GED	18 (45.0%)
2-year College	8 (20.0%)
4-year Degree	10 (25.0%)
Graduate Degree	3 (7.5%)
Health Insurance, *n* (%)	
Public	16 (35.0%)
Private	19 (47.5%)
Charity care	4 (10.0%)
I do not know	1 (2.5%)
Cancer diagnosis, *n* (%)	
Blood cancers	30 (75.0%)
Solid Tumors	10 (25.0%)
Time since diagnosis in weeks, *Md* (min-max)	7.0 (3–86)
Clinic type, *n* (%)	
Adult	33 (82.5%)
Pediatric	7 (17.5%)

**Table 2 cancers-14-03124-t002:** Participant’s Satisfaction with Bright IDEAS (*n* = 32).

	N ^a^	Strongly Disagree	Disagree	Neutral	Agree	Strongly Agree	Mean (SD)
I can see myself using the Bright IDEAS system of problem-solving	32	3.10%	0%	3.10%	46.90%	46.90%	4.34 (0.83)
After I learned the Bright IDEAS system, I thought it might be useful for friends and family	32	3.10%	0%	12.50%	28.10%	56.30%	4.34 (0.94)
I have explained Bright IDEAS to another person	32	3.10%	6.30%	21.90%	37.50%	31.30%	3.87 (1.04)
I have solved a problem using Bright IDEAS	32	0%	0%	3.10%	34.40%	62.50%	4.59 (0.46)
The Bright IDEAS program was useful ^a^	22	0%	0%	0%	40.90%	59.10%	4.59 (0.50)
The worksheets were helpful ^a^	22	0%	4.50%	13.60%	36.40%	45.50%	4.23 (0.87)
The manual explaining the program was easy to understand ^a^	22	0%	0%	13.60%	40.90%	45.50%	4.32 (0.72)
It was easy to talk with my trainer ^a^	22	0%	0%	0%	9.10%	90.90%	4.91 (0.29)

^a^ Four additional satisfaction items were added to the survey after the initial ten patients had already completed the study.

**Table 3 cancers-14-03124-t003:** Qualitative feedback about Bright IDEAS.

Prompt	Theme	Definition	Illustrative Quotes
What did you like best?	Bright IDEAS approach	Learning ways to solve problems, worksheets, ease of use, motivation to act.	“…the best that I was able to learn how to approach my problems from systematic way…” “…it made my concerns seem more manageable and less overwhelming…” “Finding new solution whenever I gotten stuck on an issue…” “easy to use and effective”
	Trainer	Someone to talk to, therapeutic	“I really enjoyed my long discussions with my trainer.” “…its like (having) a personal therapist!” “the person who listens to us…whatever we are experiencing right now.”
What did you like least?	Scheduling	Treatment schedule makes difficult to complete; clinic setting is stressful	“The space between meetings due to my treatment regimen.” “Sometimes it was during treatment when I was stressed”
	Worksheets	Homework was burdensome	“I didn’t always enjoy getting ‘homework’ assignments.” “Writing stuff down, it really did not help me. It might have helped other people. For me the worksheets and stuff was kind of juvenile. Just talking and going through it helped me.”
What could we improve?	Format	Add group/peer community component; use videocall (instead of phone)	“being able to work in group sessions” “add a group session online. People can join the group and talk to each other’s story and experience.” “Facetime might be better [than phone], its more personable, it puts face to the voice, it kinda makes it more…human.”
	Scheduling	Recruit closer to diagnosis; give more time to solve problems; add follow-up after treatment is over	“if you start it with other people, make sure they are at the beginning, because I think it will help in the beginning.” “probably given more time to solve the problems or experience them.”

**Table 4 cancers-14-03124-t004:** Changes in Secondary Psychosocial Outcomes from Baseline to Post-Intervention (*n* = 32).

	Baseline *M* (SD)	Post-Intervention *M* (SD)	Mean Difference [95% CI]	*p*	*d*
SPSI-R:S	13.90 (2.79)	15.18 (2.24)	1.28 [0.43, 2.12]	<0.01	0.54
PPO	2.71(0.83)	2.96 (0.76)	0.25 [0.05, 0.45]	0.02	0.44
NPO	2.84 (0.90)	3.08 (0.61)	0.24 [−0.02, 0.50]	0.07	0.34
RPS	2.44 (0.99)	2.91 (0.78)	0.47 [0.21, 0.73]	<0.01	0.65
ICS	2.79 (0.59)	2.86 (0.56)	0.07 [−0.13, 0.27]	0.50	0.12
AS	3.13 (0.71)	3.38 (0.52)	0.24 [−0.05, 0.54]	0.10	0.30
PROMIS—Anxiety T-score	51.98 (11.03)	49.52 (9.15)	−2.46 [−5.90, 0.99]	0.16	−0.26
PROMIS—Depression T-score	48.68 (9.62)	45.50 (8.46)	−3.18 [−6.35, −0.002]	0.05	−0.36
PedsQL Total	70.75 (16.61)	72.82 (14.81)	2.07 [−2.48, 6.62]	0.36	0.16
Distress	3.16 (2.81)	2.06 (2.16)	−1.10 [−1.96, −0.23]	0.02	−0.47

Note. SPSI-R:S: Social Problem-Solving Inventory-Revised: Short Form; PPO: Positive Problem Orientation; NPO: Negative Problem Orientation; RPS: Rational Problem-Solving Style; ICS: Impulsive/Carelessness Style; AS: Avoidance Style; PedsQL: Pediatric Quality of Life; M: mean; SD: standard deviation; d = Cohen’s d.

## Data Availability

The data that support the findings of this study are available from the corresponding author upon reasonable request.

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
