# Peer review of "Feasibility and Acceptability of Bright IDEAS-Young Adults: A Problem-Solving Skills Training Intervention"

_cancers, 2022, doi:10.3390/cancers14133124_

Round 1

Reviewer 1 Report

Thank you for the opportunity to review this paper, one that addresses an important aspect of cancer care among young adults. It is well written and concise, with appropriate detail and inclusion of tables and figures.

The following suggestions are provided in an attempt to provide clarity to the paper and increase its scope and readability.

-       The authors note that only one other intervention has been trialed in this space, but was limited to survivors under 25, as opposed to the 29-year-old cutoff in the current study. Further detail is needed as to what this intervention entailed and its effects, and what limitations prompted the adaptation and testing of the current intervention (the 4-year difference does not seem sufficiently large to exclude it from consideration).

-       Further justification could be provided as to why an intervention designed for caregivers in various contexts was the ideal model for an intervention for young adult cancer survivors. This becomes important given the dropout rates noted among certain groups (e.g., males, those with worsening disease).

-       While an issue in much psycho-oncology research, there should be brief but appropriate recognition of the demographics of those involved (e.g., primarily white, employed).

-       Could the authors include the age range of participants in the study, and subsequently comment on whether the intervention was considered appropriate/acceptable for individuals in the late teenage years as compared to those in their late 20’s.

-       The higher rate of male dropout as compared to females is worthy of comment and discussion, especially when considering possible further adaptations and potential for such an intervention to be scaled and offered more widely.

Author Response

Thank you for the opportunity to submit a revised draft of the manuscript “Feasibility and acceptability of Bright IDEAS-Young Adults: A problem-solving skills training intervention” for publication in the Special Issue "Recent Advances in Pediatric, Adolescent and Young Adult (AYA) Psycho-Oncology" in Cancers. We appreciate the thoughtful comments of the reviewers and have responded to each point below. Changes are shown  within the manuscript with tracked changes. All page numbers refer to the revised manuscript file with tracked changes.

Reviewers' Comments to the Authors:

Reviewer 1

Thank you for the opportunity to review this paper, one that addresses an important aspect of cancer care among young adults. It is well written and concise, with appropriate detail and inclusion of tables and figures.

Author response: Thank you for the kind words.

  1. The authors note that only one other intervention has been trialed in this space, but was limited to survivors under 25, as opposed to the 29-year-old cutoff in the current study. Further detail is needed as to what this intervention entailed and its effects, and what limitations prompted the adaptation and testing of the current intervention (the 4-year difference does not seem sufficiently large to exclude it from consideration).

Author response: We have added details to reflect the difference – the prior work was conducted in a pediatric setting and the majority of the sample (73%) was less than 18 years of age, thus representing the adolescent aspect of AYA and potentially limiting generalizability to adult oncology settings and older young adults.

  1. Further justification could be provided as to why an intervention designed for caregivers in various contexts was the ideal model for an intervention for young adult cancer survivors. This becomes important given the dropout rates noted among certain groups (e.g., males, those with worsening disease).

Author response: We added details on Page 2 as to why we believe this intervention model meets the needs of young adult cancer survivors.

  1. While an issue in much psycho-oncology research, there should be brief but appropriate recognition of the demographics of those involved (e.g., primarily white, employed).

Author response: We added the limitation of our sample being drawn from a single comprehensive cancer center on Page 10. We do point out, however, that our sample was relatively diverse in terms of race/ethnicity (i.e., 45% White, 42.5% Hispanic) and employment status (55% working full-time).

  1. Could the authors include the age range of participants in the study, and subsequently comment on whether the intervention was considered appropriate/acceptable for individuals in the late teenage years as compared to those in their late 20’s.

Author response: We now include the age range in Table 1 and comment on the acceptability of the intervention across the targeted age group (Page 9).

  1. The higher rate of male dropout as compared to females is worthy of comment and discussion, especially when considering possible further adaptations and potential for such an intervention to be scaled and offered more widely.

Author response: We added to the discussion regarding the higher rate of dropout among males vs. females. We are cautious in making any conclusions, however, due to our relatively small sample size and general reasons that people withdrew from the study (due to worsening health). We suggest this be monitored in future studies.

Reviewer 2 Report

Thank you for the opportunity to review “Feasibility and acceptability of Bright-IDEAS-Young Adults: A problem-solving skills training intervention”. The objective of this manuscript was to describe the feasibility and acceptability of a problem-solving intervention, originally developed for and shown efficacy among caregivers of children with cancer, that has been adapted for young adult patients with cancer. This manuscript possesses several strengths, including being well-written and addressing a clear clinical need for young adults with cancer. My main concerns center around lack of specify about how this intervention was adapted for young adults (or was the same intervention originally developed for caregivers delivered to young adults?) and clarity of methods for this pilot trial (including progression criteria for an RCT). 

1)    Did the authors have pre-established thresholds for determining feasibility and acceptability of the adapted intervention (e.g., what recruitment rate would correspond to “adequate”?) Were their apriori hypotheses? How would you know results reached progression criteria for full-scale RCT? 

2)    Related to my comment above, based on the CONSORT diagram, a little over half of approached, eligible young adults completed the pilot trial. This could indicate feasibility challenges that warrant more explicit discussion about ways the intervention may be optimized for this population in the future. 

3)    It was unclear if or how the original Bright-IDEAS intervention changed for young adults (e.g., to the intervention content, mode of intervention delivery, number of sessions), or if the research team tested the same original intervention (designed for caregivers) in a young adult population. If the intervention was not modified in any way, why not? Is there any empirical data (e.g., qualitative data) to suggest that young adults did not want or need any changes to the original intervention? Some of the feasibility and acceptability data made me wonder if specific adaptations were needed for this population (e.g., the desire for a peer component, greater flexibility with scheduling sessions).

4)    On pg. 3 (measures), it would be helpful to delineate metrics of feasibility more clearly (e.g., recruitment, retention) and measures of acceptability (e.g., perceived satisfaction and usefulness of the intervention).

5)    Were there any checks of intervention fidelity? 

6)    How soon after the intervention ended did participants complete the post-questionnaires?

7)    Rationale for including young adults aged 18-29?

8)    How did the chosen “problems” that young adults selected for the intervention compare to caregivers in previous Bright-IDEAS trials?   

9)    Did the authors explore whether any of the heterogeneity described as limitations (e.g., diagnosis type, time since diagnosis, or other factors such as age, race, ethnicity, or pediatric vs. adult) related to core feasibility and acceptability outcomes? 

10) The authors state in the discussion that they were not adequately powered for exploratory analyses, but do not present a formal power analysis within the paper. 

11) Some of the unmet psychosocial needs faced by young adults with cancer that the authors described in the Introduction (particularly “practical” needs such as insurance and transportation) were not specifically targeted by the intervention or measured in this study. These needs might also reflect systemic changes that are needed (particularly for minoritized populations), as opposed to individual-level interventions. It might be useful to temper this background and focus most on the immediate targets of the intervention (problem-solving skills, symptoms of depression and anxiety), then address more specifically how problem-solving can be flexibly applied to a range of problems that young adults with cancer face. 

Author Response

Reviewer 2:

  1. Did the authors have pre-established thresholds for determining feasibility and acceptability of the adapted intervention (e.g., what recruitment rate would correspond to “adequate”?) Were their apriori hypotheses? How would you know results reached progression criteria for full-scale RCT?

Author response: We have clarified that we set out to determine rates of enrollment, withdrawal, etc. using guiding questions as suggested in Osmond and Cohn (2015), which we now cite in the Introduction. We did not have a priori hypotheses and have now stated this clearly in the Discussion. We also now compare our rates to a similar published study.

  1. Related to my comment above, based on the CONSORT diagram, a little over half of approached, eligible young adults completed the pilot trial. This could indicate feasibility challenges that warrant more explicit discussion about ways the intervention may be optimized for this population in the future.

Author response: We believe our enrollment rate (67.8%) is comparable to other similar studies; for example, Rosenberg et al. 2018 pilot study resulted in a similar enrollment rate at 71%. However, we agree that a discussion on recruitment and retention is warranted, especially for this population. We have added to the Discussion.

  1. It was unclear if or how the original Bright-IDEAS intervention changed for young adults (e.g., to the intervention content, mode of intervention delivery, number of sessions), or if the research team tested the same original intervention (designed for caregivers) in a young adult population. If the intervention was not modified in any way, why not? Is there any empirical data (e.g., qualitative data) to suggest that young adults did not want or need any changes to the original intervention? Some of the feasibility and acceptability data made me wonder if specific adaptations were needed for this population (e.g., the desire for a peer component, greater flexibility with scheduling sessions).

Author response: We welcome the opportunity to add these details. We have now noted in section 2.2 how we adapted the intervention for young adults. Specifically, we modified Bright IDEAS-YA to be six sessions, which was determined in prior studies to be an adequate dose to achieve effects. We also changed the participant materials to be simpler/more appealing, added examples relevant to young adults (e.g., changing a caregiver example of having questions about their child’s diagnosis to a young adult having questions about their own diagnosis), added psychoeducation regarding changeable vs. unchangeable aspects of problems, and expand the use of word “problems” to include “problems,” “challenges,” or “goals” to capture the range of issues young adults might experience.

  1. On pg. 3 (measures), it would be helpful to delineate metrics of feasibility more clearly (e.g., recruitment, retention) and measures of acceptability (e.g., perceived satisfaction and usefulness of the intervention).

Author response: We revised as suggested.

  1. Were there any checks of intervention fidelity?

Author response: Yes, we audio recorded all sessions (with patient permission) and used the recordings for supervision and treatment integrity assessments. This is added to section 2.2.

  1. How soon after the intervention ended did participants complete the post-questionnaires?

Author response: Participants completed the post-intervention survey immediately following the final intervention session. We clarified this on p. 3 in section 2.3.

  1. Rationale for including young adults aged 18-29.

Author response: The selection of ages 18-29 is based on the definition of emerging adulthood and the unique emotional, social, and practical needs during this time. A cancer diagnosis makes this group particularly vulnerable due to the intersection of cancer treatment and normal developmental processes. We have added a sentence to the Introduction to provide this rationale.

  1. How did the chosen “problems” that young adults selected for the intervention compare to caregivers in previous Bright-IDEAS trials.

Author response: We did not compare the problems discussed by young adults directly to the caregiver trials. However, in the Discussion, we note that the types of problems discussed reflect the problems noted in the literature on young adult cancer survivors. “Consistent with the broader literature,33, 34 young adults in our study reported a number of challenges during treatment for cancer, including coping with the uncertainty of cancer, challenges with work/careers or returning to school, social difficulties with friends or romantic relationships, engaging in healthy behaviors, and losing autonomy due to moving back under the care of parents.”

  1. Did the authors explore whether any of the heterogeneity described as limitations (e.g., diagnosis type, time since diagnosis, or other factors such as age, race, ethnicity, or pediatric vs. adult) related to core feasibility and acceptability outcomes.

Author response: Our results suggest that there were no significant differences between participants and non-participants in age, sex, race, ethnicity, diagnosis type (blood cancer vs. other), or time since diagnosis. In terms of dropout, only gender (male) and enrolled at a longer time from the initial diagnosis were more likely to withdraw; there were no other significant differences among other demographic characteristics. Lastly, we did not examine acceptability/satisfaction by demographics due to the small number of individuals providing negative ratings (i.e., only 1 or 2 individuals giving negative ratings across items).

  1. The authors state in the discussion that they were not adequately powered for exploratory analyses, but do not present a formal power analysis within the paper.

Author response: Since this study focuses on feasibility and determining acceptability of the adapted Bright IDEAS intervention among young adults with cancer, we did not conduct an a priori power calculation regarding outcome effects. Given the exploratory nature of the outcome analyses, we focus on discussion of the size of effects rather than their statistical significance. We believe this is well-aligned with the purpose of feasibility studies and have added to both the Introduction and Discussion to clarify these goals.

  1. Some of the unmet psychosocial needs faced by young adults with cancer that the authors described in the Introduction (particularly “practical” needs such as insurance and transportation) were not specifically targeted by the intervention or measured in this study. These needs might also reflect systemic changes that are needed (particularly for minoritized populations), as opposed to individual-level interventions. It might be useful to temper this background and focus most on the immediate targets of the intervention (problem-solving skills, symptoms of depression and anxiety), then address more specifically how problem-solving can be flexibly applied to a range of problems that young adults with cancer face.

Author response: As suggested, we tempered the Introduction by removing some of the details regarding unmet needs, identified individual interventions as one possible way forward, and then focused on the immediate targets of the intervention. Bright IDEAS is an individual-level intervention where the focus is on teaching the participant better problem-solving skills that they can apply to any problem of their choosing (i.e., practical, emotional, social, etc.).

Round 2

Reviewer 2 Report

The authors have been very responsive to the prior review. I have no further suggestions and I look forward to seeing this paper published!